# Cotreatment of MSWI Fly Ash and Granulated Lead Smelting Slag Using a Geopolymer System

**DOI:** 10.3390/ijerph16010156

**Published:** 2019-01-08

**Authors:** De-Gang Liu, Yong Ke, Xiao-Bo Min, Yan-Jie Liang, Zhong-Bing Wang, Yuan-Cheng Li, Jiang-Chi Fei, Li-Wei Yao, Hui Xu, Guang-Hua Jiang

**Affiliations:** 1Institute of Environmental Science and Engineering, School of Metallurgy and Environment, Central South University, Changsha 410083, Hunan, China; liudegang6669@163.com (D.-G.L.); keyong000ke@csu.edu.cn (Y.K.); csulyj555@163.com (Y.-J.L.); wzbcsu@163.com (Z.-B.W.); li245547278@hotmail.com (Y.-C.L.); jack-fei@csu.edu.cn (J.-C.F.); yaoliwei0125@126.com (L.-W.Y.); xuhuiyh@csu.edu.cn (H.X.); 15073209975@163.com (G.-H.J.); 2Chinese National Engineering Research Center for Control and Treatment of Heavy Metal Pollution, Changsha 410083, Hunan, China

**Keywords:** MSWI fly ash, granulated lead smelting slag, geopolymer, solidification/stabilization, heavy metals, Friedel’s salt

## Abstract

Municipal solid waste incineration fly ash (MSWI FA) and granulated lead smelting slag (GLSS) are toxic industrial wastes. In the present study, granulated lead smelting slag (GLSS) was pretreated as a geopolymer precursor through the high-energy ball milling activation process, which could be used as a geopolymeric solidification/stabilization (S/S) reagent for MSWI FA. The S/S process has been estimated through the physical properties and heavy metals leachability of the S/S matrices. The results show that the compressive strength of the geopolymer matrix reaches 15.32 MPa after curing for 28 days under the best parameters, and the physical properties meet the requirement of MU10 grade fly ash brick. In addition, the toxicity characteristic leaching procedure (TCLP) test results show that arsenic and heavy metals are immobilized effectively in the geopolymer matrix, and their concentrations in the leachate are far below the US EPA TCLP limits. The hydration products of the geopolymer binder are characterized by X-ray diffraction and Fourier transform infrared methods. The results show that the geopolymer gel and Friedel’s salt are the main hydration products. The S/S mechanism of the arsenic and heavy metals in the geopolymer matrix mainly involves physical encapsulation of the geopolymer gel, geopolymer adsorption and ion exchange of Friedel’s salt.

## 1. Introduction

The amount of municipal solid waste (MSW) produced in China was approximately 203 million tons in 2016 [1]. Due to the large quantity of MSW generated annually and the shortage of landfill disposal, incineration technology is becoming the first choice for MSW disposal. Between 2004 and 2016, the quantity of incinerated MSW increased from 4.49 to 73.78 million tons annually, and the number of MSW incineration plants increased from 54 to 249 in China [1,2]. Although MSW incineration can produce abundant energy and decrease the waste volume by 85–90% [3], it still has some disadvantages. First, the potentially hazardous municipal solid waste incineration fly ash (MSWI FA) is produced, and the weight of MSWI FA is up to 2–5 wt.% of the initial MSW [4]. Moreover, MSWI FA contains poisonous contaminants, such as heavy metals, dioxins, and furan. As a result, MSWI FA is identified as a hazardous waste according to the List of Hazardous Waste in China. Therefore, MSWI FA should be disposed before the final treatment for decreasing the leaching concentration of the contaminants. In general, the treatments of MSWI FA are mainly divided into three types, that is, separation, thermal treatment, and solidification/stabilization (S/S) [5]. To date, the S/S technology with ordinary Portland cement (OPC) has been used to treat MSWI FA in many countries for several years [6,7,8]. Nevertheless, the disposal of MSWI FA with OPC has become increasingly difficult at the present, owing to the high cost and frequent public opposition to the sifting of new landfills [9]. Due to those reasons, methods that can solidify/stabilize MSWI FA with another waste are urgently needed.

Geopolymer binder is produced from materials containing aluminosilicate (e.g., metakaolin, kaolinite [10], coal fly ash [11], red mud [12] and blast furnace slag [13], etc.) with alkali-activation via the dissolution-polycondensation process [14]. Geopolymer binder possesses a better durability and leads to lower CO_2_ emission than OPC. Hence, geopolymer as an innovative binder for S/S technology has been utilized to dispose MSWI FA in recent years. For example, Lancellotti et al. [15] utilized metakaolin as the main raw material to synthesize geopolymer for S/S of MSWI FA. Zheng et al. [16] investigated the synthesis of MSWI FA-based geopolymer with water glass after water-wash pretreatment for immobilizing heavy metals. Some researchers [17,18] also investigated the codisposal of MSWI FA and coal fly ash (or red mud) for immobilization of their heavy metals. However, those kinds of treatment methods are either costly, or the MSWI FA-based geopolymer exhibits a low compressive strength when the content and molar ratio of SiO_2_ and Al_2_O_3_ are low. Thus, improving the compressive strength of this kind of MSWI FA-based geopolymer with another industrial waste is very meaningful. According to the previous studies [19,20], improving the content of SiO_2_ and Al_2_O_3_, and the molar ratio of SiO_2_ and Al_2_O_3_ up to 3.4–4.5 can remarkably increase the compressive strength of the geopolymer binder. To our knowledge, granulated lead smelting slag (GLSS) contains plenty of SiO_2_ and Al_2_O_3_, and the molar ratio of SiO_2_ and Al_2_O_3_ is high, which can be considered as an aluminosilicate source for geopolymer production.

GLSS is an industrial waste generated from the smelting furnace during the lead smelting concentration after the smelting process from primary lead production and is rich in Fe, Ca, Si, Al and Zn [21,22]. GLSS is also a toxic industrial waste since it contains quantities of minor and trace heavy metals elements [23,24,25,26,27], and it can contaminate the environment through leaching if it is not constrained [28,29,30,31]. According to the statistics, approximately 2.56 Mt of lead smelting slag is being generated annually in China [32], imposing a great burden on environment protection. As a consequence, arsenic and heavy metals of GLSS and MSWI FA need to be immobilized effectively. According to the previous study, the anionic species is often difficult to solidify/stabilize in geopolymers, but Friedel’s salt seems to immobilize them effectively. Recently, Friedel’s salt has attracted increasing attention in the past several years because of its good ion exchange characteristics [33,34,35,36,37]. It was reported that Friedel’s salt could effectively adsorb Zn^2+^ [38], AsO_4_^3−^ and PbO_2_^−^ from wastewater. In addition, MSWI FA and GLSS may synthesize Friedel’s salt with alkali activation. Therefore, Friedel’s salt may immobilize arsenic and heavy metals effectively in the geopolymer binder. Moreover, Shi and Day [39] reported that Friedel’s salt can improve the compressive strength of the slag-based binder, and Cheng et al. [40] also reported that Friedel’s salt may be correlated to the strength enhancement of the S/S matrix. Due to the interesting findings of these studies, the possibility of cotreatment of MSWI FA and GLSS using geopolymerization was investigated. Although this paper focuses on the immobilization of arsenic and heavy metals in the geopolymer binder, it can also be utilized as a replacement for civil engineering materials in the future due to its good physical properties.

The aim of the research was to immobilize MSWI FA using another industrial waste, i.e., GLSS, through the geopolymerization system. First, the reaction products of the geopolymer binder were characterized. Second, the performances tests (i.e., compressive strength, water absorption, dry shrinkage, dry density and heavy metals leachability) of the geopolymer matrix were performed. Finally, the S/S mechanism of arsenic and heavy metals in the geopolymer matrix was discussed.

## 2. Materials and Methods

### 2.1. Materials

The MSWI FA was sampled from a waste incineration plant located in the south of China. The MSWI FA was dried at 105 °C. The particle size distribution of the MSWI FA powder is presented in Figure 1. It is obvious that 90% of the MSWI FA particles possess a spherical equivalent diameter below 100 µm, whereas 50% is below 34 µm, and 10% is below 6 µm. Therefore, the MSWI FA did not require grinding pretreatment. The GLSS used in this study was collected from a lead smelter in the south of China. The GLSS was dried at 105 °C in a drying cabinet until a constant mass was reached. Then, the GLSS was ground in a planetary grinding machine for 3 h and passed through a 45 µm mesh sieve [41]. The particle size of the GLSS powder was determined using laser granulometry. As shown in Figure 1, the particle size of the GLSS powder is in the range of 0.92–41.84 µm with a D_50_ of 5.28 µm.

The bulk chemical analysis results of the MSWI FA and GLSS presented in Table 1 were obtained using a Perkin Elmer ICP-OES following the digestion of a 0.5 g sample. In addition, Cl and Si were analyzed by X-ray fluorescence (XRF). As shown in Table 1, the main elements of the MSWI FA are Ca (28.7%), Cl (17.98%), K (2.57%), Si (2.62%) and Al (2.49%), while the major elements of the GLSS are Fe (26.9%), Si (15.93%), Ca (9.55%), Al (4.32%) and Zn (2.81%). As shown in Figure 2, the GLSS was mainly consisted of an amorphous phase and some crystalline phases of magnetite. The broad and diffuse peaks from the GLSS approximately 28–35° (2*θ*) reflected the short-range order of the CaO-Al_2_O_3_-MgO-Fe_x_O-SiO_2_ glass structure. This is a common feature of the amorphous phase, which is an indication of a rather reactive phase [42]. The leachability of heavy metals was measured using the toxicity characteristic leaching procedure (TCLP) method. According to Table 2, the concentrations of Pb (8.47 mg/L) and Zn (167.16 mg/L) were found to be high in the MSWI FA and GLSS respectively, exceeding the limits of the regulations in China.

The sodium silicate was supplied by Shandong Usolf Chemical Technology Co., Ltd. (LinYi, China). The product specifications of the sodium silicate are listed as follows: SiO_2_ (26.5 wt.%), Na_2_O (8.3 wt.%), molar ratio of SiO_2_/Na_2_O of 3.3 and density of 1.371 g/cm^3^. The other chemicals were of analytical grade.

### 2.2. Experimental Procedures

The MSWI FA was mixed with the GLSS at percentages of 0% (L0M100), 20% (L20M80), 40% (L40M60), 60% (L60M40) and 80% (L80M20), respectively. The mixture samples, sodium silicate and deionized water were added to obtain homogeneous pastes in a paste mixer. Then, the pastes were poured into steel molds (20 mm × 20 mm × 20 mm) and were covered with plastic films. Afterwards, they were cured for 24 h in a cement concrete standard curing box at 20 ± 2 °C and 95% relative humidity. The matrices were demolded and were cured again under the same conditions. Then, the unconfined compressive strength (UCS) and TCLP tests of the geopolymer matrices were performed after curing for 3, 7 and 28 days.

### 2.3. Tests

The XRF test can provide a qualitative identification and quantitative analysis of elements for solid samples. The MSWI FA and GLSS samples were initially ground to make sure the particle size was less than 45 µm and then pressed to a pellet. An XRF analyzer (S4-Pioneer, Bruker Ltd., Karlsruhe, Germany) then was used for the quantitative chemical analysis of the samples.

X-ray diffraction (XRD) was utilized to determine the mineral compositions of MSWI FA, GLSS, and geopolymer matrices. The samples were ground into powders with a particle size less than 45 µm as a pretreatment. The crystal compositions of the samples were characterized by an XRD tester (D/max2550 VB + 18 KW) with Cu Kα radiation at a scanning rate of 10°/min from 5° to 65°. The diffraction patterns were manually analyzed using the Joint Committee on Powder Diffraction standards.

The infrared spectra of the MSWI FA and the geopolymer matrices were obtained with a Fourier transform infrared (FTIR) spectrometer (Nicolet IS10, Thermo Fisher Scientific, Waltham, MA, USA), and the samples were prepared using the standard KBr pellet method.

The effectiveness of S/S should be characterized for determining the environmental impact of the treated waste before it is disposed or reused [43]. The TCLP test is used to estimate the amount of contaminants that might be released to the environment in simulated landfill conditions, and the concentrations of contaminants in the leachate [44]. The samples were crushed to less than 0.95 cm manually. Then, the crushed samples were leached in an extraction buffer, with acetic acid (pH 2.88 ± 0.05) at a liquid/solid ratio of 20:1. The extraction (at 25 ± 2 °C) was performed by shaking on a shaker at a speed of 30 rpm for 18 h. Subsequently, the leachates were filtered with a 0.45 µm fiber filter, and then the arsenic and heavy metals concentrations of the filtrates were analyzed using ICP-AES (IRIS Intrepid II XSP, Thermo Electron Corporation, Waltham, MA, USA).

## 3. Results and Discussion

### 3.1. Characterization of the Reaction Products in the Geopolymer Binder

To understand what were responsible for the physical properties and heavy metals immobilization of the geopolymer matrix, the reaction products could be characterized using XRD and FTIR methods. The MSWI FA and GLSS samples, as well as selected geopolymer matrices, were examined by XRD. As shown in Figure 2, the main crystalline phase of the MSWI FA was identified as CaClOH, SiO_2_, NaCl, KCl, Ca(OH)_2_ and CaCO_3_. The results agreed with that of Kougemitrou et al. [45,46]. When the MSWI FA was alkali-activated without GLSS addition (L0M100), the peak of CaClOH disappeared. At the same time, the peak intensities of CaCO_3_ and NaCl increased, and a new weak peak of Friedel’s salt appeared. In addition, the geopolymer gel could be observed, which was attributed to the increase of the amorphous “hump” between 20° and 40° according to Phair and van Deventer [47]. Therefore, the following reactions may have occurred:CaClOH + NaOH → Ca(OH)_2_ + NaCl(1)
Ca(OH)_2_ + CO_2_ → CaCO_3_ + H_2_O(2)
M + Si(OH)_4_^−^ + Al(OH)_4_^−^ → M_n_-[-(SiO_2_)_z_-AlO_2_]_n_∙wH_2_O (Geopolymer)(3)
Ca^2+^ + Al(OH)_4_^−^ + Cl^−^ + H_2_O → 3CaO∙Al_2_O_3_∙CaCl_2_∙10H_2_O (Friedel’s salt)(4)
where M is a cation, usually a Na^+^, K^+^ or Ca^2+^, n is a degree of polycondensation, w ≤ 3 and z is 1, 2 or 3.

When the GLSS was added to the MSWI FA, the XRD test results of L40M60 were similar to that of L0M100 except that the peak intensities of Friedel’s salt increased obviously. The result indicated that Al_2_O_3_ was dissolved from the GLSS under alkaline conditions, and then generated more Friedel’s salt. However, when the dosage of GLSS was increased to 80% (L80M20), the peaks of Friedel’s salt were not detected, indicating that the content of the Friedel’s salt is too low to detect in the geopolymer matrix.

To verify the XRD results of the geopolymer binder during the hydration process, the microstructure of the original MSWI FA and geopolymer matrix was characterized by FTIR. The FTIR spectra of the original MSWI FA and the geopolymer matrices (L0M100 and L40M60) are shown in Figure 3. The bands located at 3570 and 3643 cm^−1^ of the original MSWI FA were attributed to the asymmetric stretching vibration of -O-H in Ca(OH)_2_, and they disappeared in L0M100 and L40M60, indicating that the Ca(OH)_2_ reacted with others during the hydration process. The bands located at 1422–1471 cm^−1^ of the original MSWI FA, L0M100 and L40M60 are attributed to the O-C-O asymmetric stretching in the samples [48]; those bands implied the presence of carbonate. The band located at 1155 cm^−1^ of the original MSWI FA is attributed to the Si-O-Si asymmetric stretching vibration [49], which shifts to higher energies in L0M100 (1157 cm^−1^) and L40M60 (1157 cm^−1^). This was indicative of a higher degree of the silicate networks after geopolymerization. The band located at 984 cm^−1^ in the original MSWI FA was assigned to Al-O-Si, which is associated with the Si/Al molar ratio in the geopolymer framework, and can shift to a higher wavenumber at a higher molar ratio of Si/Al [50]. Hence, the band peak in the original MSWI FA shifted to a higher number (998 cm^−1^) in L0M100. The band located at 679 cm^−1^ in the original MSWI FA was assigned to the amorphous aluminosilicate containing ring structures [48]. The amorphous aluminosilicate was dissolved and the band peak disappeared after geopolymerization. According to the pure Friedel’s salt phase, there is a very strong feature at approximately 532 cm^−1^ [51] and 786 cm^−1^ [52] which can be ascribed to the Al-OH bending mode. This band shifted to a higher frequency of 537 cm^−1^ and 789 cm^−1^ for both L0M100 and L40M60, which possibly meant that the arsenic or heavy metals intruded into the structure of Friedel’s salt.

### 3.2. Compressive Strength

#### 3.2.1. Effect of GLSS Addition

The UCS of the geopolymer matrices with different dosages of GLSS is shown in Figure 4. With the GLSS dosage increasing from 0 to 40%, the UCS of the geopolymer matrices increased after curing for 3, 7 and 28 days, and then decreased with the further increase of the dosage of GLSS from 40% to 80%. The geopolymer matrix with the GLSS dosage of 40% showed a high UCS of 10.36 MPa at the curing age of 3 days and reached the highest value of 12.87 MPa at 28 days. Conclusively, the optimal GLSS dosage of 40% was selected. This increase was due to the increase of the molar ratio of SiO_2_/Al_2_O_3_ from 2.1 to 3.8 in the geopolymer matrices.

According to Khale et al. [19,20], increasing the molar ratio of SiO_2_/Al_2_O_3_ up to 3.4–4.5 is greatly responsible for the high long term strength of the geopolymer matrices. Moreover, the UCS increase is also due to the bonds of Si-O-Si becoming stronger, when the geopolymer matrix is enriched with a greater amount of Si [53]. In addition, chloride could significantly retard the solidification of the geopolymer gel [54], and lower the UCS by causing structural discontinuity within the geopolymer gel [55]. Ferone et al. [56] have also demonstrated the negative effect of the presence of chloride on the polycondensation kinetic. However, Ca(OH)_2_, Cl^−^ and Al_2_O_3_ are conducive to form Friedel’s salt (Equation (4)), which can remarkably reduce the concentration of free chloride ions in the geopolymer binder. In addition, the formation of Friedel’s salt can improve the UCS by filling in the pores of the S/S matrix [39,40]. It could explain the reason why the UCS of the geopolymer matrices reached the highest when the GLSS dosage was 40%.

#### 3.2.2. Effect of Modulus and Dosage of Sodium Silicate

The UCS of the geopolymer matrices with different modulus and dosages of sodium silicate is shown in Figure 5. With the increasing dosage of sodium silicate, the UCS of the geopolymer matrix increased and then decreased remarkably. On the one hand, the increased dosage of sodium silicate can accelerate the dissolution of Al and Si from the MSWI FA and GLSS, and then the hydration rate increased. On the other hand, the sodium silicate provides soluble Si, which is essential for producing geopolymer gels [57]. However, the excess soluble Si would cause cracking on the surface of the geopolymer matrix according to previous studies. As a result, the optimum dosage of sodium silicate W_Na2O+SiO2_ is 5% and the modulus is 1.5. Under these conditions, the UCS of the geopolymer matrix reached the highest value (15.32 MPa) after curing for 28 days.

### 3.3. Other Physical Properties

To examine the possible usage of the geopolymer binder as a potential substitute for fly ash bricks, the performances of the geopolymer binder were evaluated based on the building materials industry standard in China (JC239-2001). According to the criteria, the UCS of MU10 grade fly ash brick is commonly between 10 and 15 MPa, and the UCS of the optimal geopolymer matrix was approximately 15 MPa after curing for 28 days. This result showed that the optimal geopolymer matrix met the UCS requirement of MU10 grade fly ash brick. Water absorption is also a significant parameter of durability for fly ash brick. The water absorption is commonly between 14.29% and 16.70% for bricks presented in the previous literature [58], and the water absorption of the optimal geopolymer matrix was 14.44%. The dry shrinkage of fly ash brick is usually between 0.040% and 0.075% according to the criteria. The dry shrinkage and dry density of the optimal geopolymer matrix was 0.058% and 1.67 g/cm^3^, respectively. From what has been discussed above, a conclusion may be safely drawn that the cotreatment of MSWI FA and GLSS with alkali-activation provided a possible method to utilize them as bricks.

### 3.4. Heavy Metals Leachability

The TCLP test results of the geopolymer matrices after curing for 28 days and the corresponding limits of waste landfills are given in Table 3. As seen, the results showed that the Pb, As, Cd, Ni, Cr, Ba and Cu leaching concentrations of the geopolymer matrices with GLSS addition were far below the limits allowed, and decreased with the increasing dosage of GLSS except Cd, Cr, Cu and Ni. In addition, the Zn leaching concentrations of GLSS and MSWI FA were 167.16 mg/L and 0.42 mg/L, respectively, but they have fallen to <0.01 mg/L in all geopolymer matrices except L80M20 (3.3 mg/L). The Ag, Hg, Be and Se leaching concentrations of the geopolymer matrices were still at <0.01 mg/L. Meanwhile, from the results of XRD and FTIR analyses, geopolymer gel and Friedel’s salt are the main phases in the geopolymer matrices. Therefore, it can be deduced that the immobilization of arsenic and heavy metals in matrix had a great relation with Friedel’s salt and the geopolymer gel.

Based on the discussion above, the S/S of MSWI FA and GLSS into geopolymers is an effective way to immobilize heavy metals in both of them. Actually, the S/S mechanisms of heavy metals mainly involve physical encapsulation, adsorption, ion exchange, and precipitation of stable compounds in the binders [59]. According to the results of the TCLP tests, Zn, Pb, As, Cd, Ni, Cr, Ba and Cu in the MSWI FA and GLSS can be solidified/stabilized effectively. In terms of the UCS, the values of the geopolymer matrices were high, so it can be deduced that physical encapsulation is responsible for fixing heavy metals in the geopolymer matrices. The geopolymer is also an important phase for stabilizing Zn^2+^, Pb^2+^, Cd^2+^, Ni^2+^ and other heavy metals through adsorption [60,61]. Moreover, Friedel’s salt has a strong fixing capacity for arsenic and heavy metals, not only the anions of AsO_4_^3−^ [62], CrO_4_^2−^ and PbO_2_^−^ [63] but also the cations of Zn^2+^ [38] and Cd^2+^ [64]. For instance, Zhang et al. discovered that Friedel’s salt can fix AsO_4_^3−^ effectively through anion exchange [65]. Additionally, Zn^2+^ can be immobilized via the primary precipitation of Zn-Friedel’s salt (3ZnO∙Al_2_O_3_∙ZnCl_2_∙10H_2_O) in a solution medium as reported by Liu et al. [38]. As a result, the S/S mechanism of arsenic and heavy metals in the geopolymer matrix mainly involves the physical encapsulation of the geopolymer gel, geopolymer adsorption, and ion exchange of Friedel’s salt.

## 4. Conclusions

This study demonstrates that MSWI FA is successfully cotreated with GLSS. The UCS of the geopolymer matrices increases significantly with the GLSS addition. As a result, the UCS of the geopolymer matrix has reached 15.32 MPa after curing for 28 days, under the best parameters of this binder (mass ratio of m(GLSS): m(MSWI FA) is 40:60, modulus of sodium silicate is 1.5, and dosage of sodium silicate (W_Na2O+SiO2_ is 5%).

Furthermore, the measured arsenic and heavy metals from all geopolymer matrices with the GLSS addition are far below the permitted limits of the US EPA. The S/S mechanism of the arsenic and heavy metals in the geopolymer matrix mainly involves physical encapsulation of the geopolymer gel, geopolymer adsorption, and ion exchange of Friedel’s salt. Additionally, the physical properties of the optimal geopolymer matrix meet the performance requirements of MU10 grade of fly ash brick. Therefore, the cotreatment of MSWI FA and GLSS using the geopolymer system has provided a potential method to utilize those two wastes as civil engineering materials. However, the long-time durability and the chloride leaching behavior of the geopolymer matrix should be studied in depth in the future.

## Figures and Tables

**Figure 1 ijerph-16-00156-f001:**
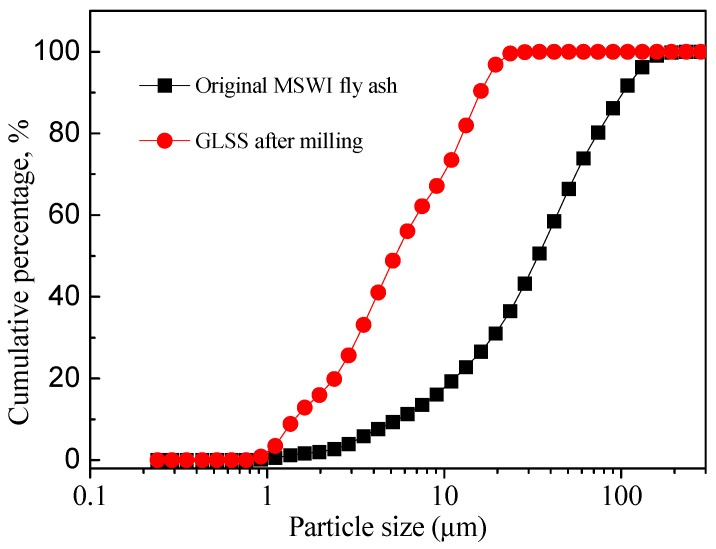
Particle size distributions of the original MSWI FA and GLSS after milling.

**Figure 2 ijerph-16-00156-f002:**
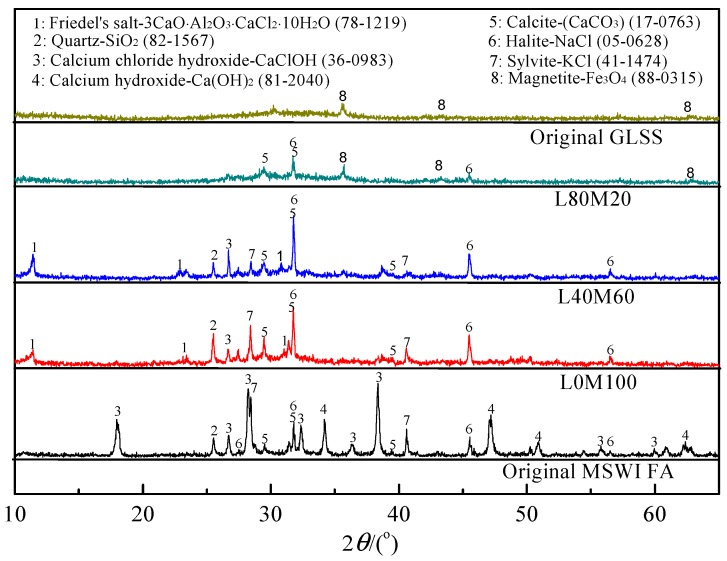
XRD patterns of the MSWI FA, GLSS, and geopolymer matrices.

**Figure 3 ijerph-16-00156-f003:**
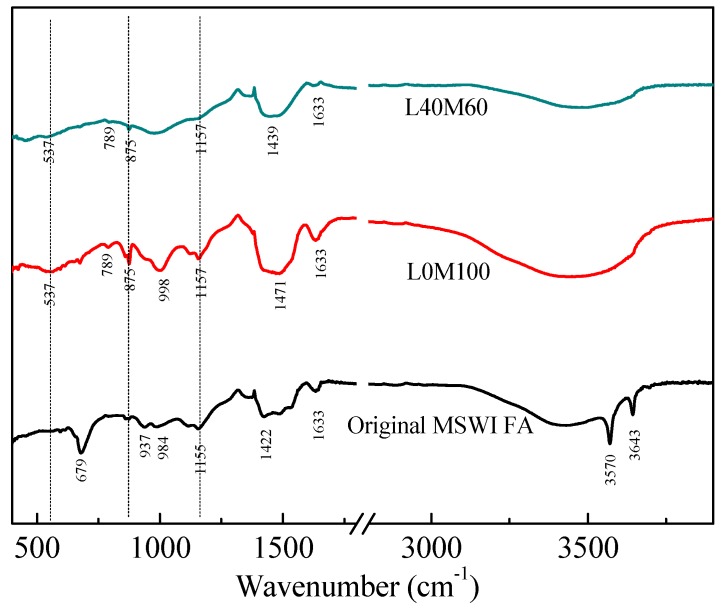
FTIR patterns of the original MSWI FA and geopolymer matrices.

**Figure 4 ijerph-16-00156-f004:**
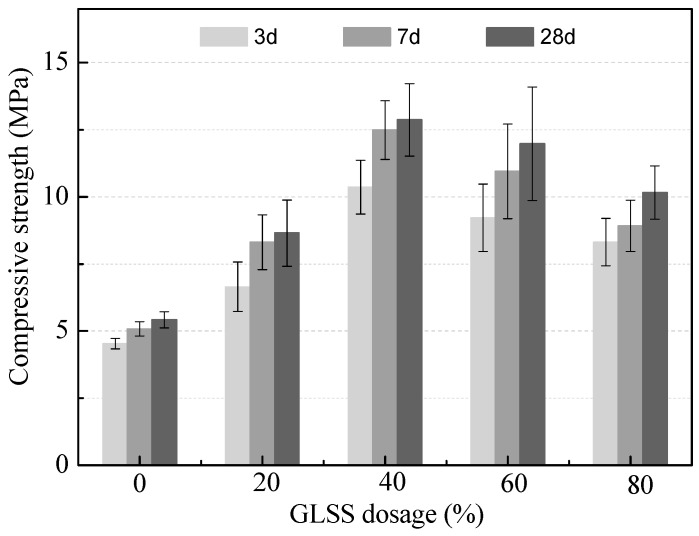
The UCS variations of the geopolymer binder with GLSS addition.

**Figure 5 ijerph-16-00156-f005:**
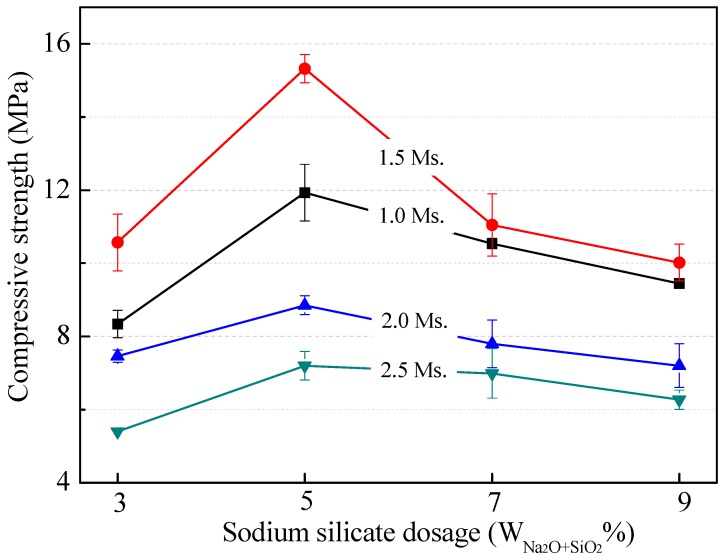
The UCS variation of the geopolymer matrices with different modulus and dosages of sodium silicate.

**Table 1 ijerph-16-00156-t001:** Main elemental compositions of the raw materials (wt.%).

**Element**	**Ag**	**Al**	**As**	**Ba**	**Be**	**Ca**	**Cd**	**Cr**	**Cu**	**Fe**	**Hg**
MSWI FA	0.0005	2.49	0.030	0.087	<0.0001	28.7	0.017	0.016	0.049	1.32	<0.0005
GLSS	0.0007	4.32	0.011	0.49	0.0002	9.55	0.0025	0.14	0.19	26.9	<0.0005
**Element**	**K**	**Mg**	**Mn**	**Na**	**Ni**	**Pb**	**S**	**Zn**	**Se**	**Si**	**Cl**
MSWI FA	2.57	1.39	0.053	1.73	0.0056	0.16	2.60	0.57	0.0016	2.62	17.98
GLSS	0.95	1.18	1.44	0.51	0.035	0.20	0.22	2.81	0.0018	15.93	-

**Table 2 ijerph-16-00156-t002:** TCLP leaching values of the GLSS and MSWI FA samples.

Element	Zn	Pb	As	Cd	Ni	Cr	Ba	Cu	Ag	Hg	Be	Se
USEPA Limits	-	5	5	1	-	5	100	100	-	0.2	-	-
GLSS	167.16	0.15	0.05	0.22	0.45	0.01	12.53	0.07	<0.01	<0.01	<0.01	<0.01
MSWI FA	0.42	8.47	0.63	<0.01	<0.01	0.09	1.94	<0.01	<0.01	<0.01	<0.01	<0.01

**Table 3 ijerph-16-00156-t003:** TCLP test results of the geopolymer matrices cured for 28 days.

Element	Zn	Pb	As	Cd	Ni	Cr	Ba	Cu	Ag	Hg	Be	Se
US EPA Limits	-	5	5	1	-	5	100	100	-	0.2	-	-
L0M100	<0.01	2.16	0.12	<0.01	<0.01	0.02	0.85	<0.01	<0.01	<0.01	<0.01	<0.01
L20M80	<0.01	0.25	0.12	<0.01	<0.01	0.02	0.68	<0.01	<0.01	<0.01	<0.01	<0.01
L40M60	<0.01	<0.01	0.10	<0.01	<0.01	0.03	0.59	<0.01	<0.01	<0.01	<0.01	<0.01
L60M40	<0.01	<0.01	<0.01	<0.01	<0.01	<0.01	0.42	<0.01	<0.01	<0.01	<0.01	<0.01
L80M20	3.30	<0.01	<0.01	0.1	0.1	<0.01	0.45	<0.01	<0.01	<0.01	<0.01	<0.01
Optimal matrix	<0.01	<0.01	0.12	<0.01	<0.01	<0.01	<0.01	<0.01	<0.01	<0.01	<0.01	<0.01

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
