# Peer review of "Cotreatment of MSWI Fly Ash and Granulated Lead Smelting Slag Using a Geopolymer System"

_ijerph, 2019, doi:10.3390/ijerph16010156_

Round 1
Reviewer 1 Report
1. The topic is of general interest, and the presentation is clear.
2. This article contains new aspects regarding the immobilization of heavy metals.
3. In manuscript the authors underline the major findings of their work and explain how the use of their proposed procedures and materials represents a a progress comparatively with other researches.
4. In Abstract please rewrite the phrases: “and the other physical properties meet….” and “and the immobilization of Zn; Pb; As; Ni and Cd has a great relation with them”.
5. The key words permit found article in the current registers or indexes.
6. The Introduction reflects state of the art.
7. The text is easy to understand by scientists in other disciplines.
8. The text is presented and arranged clearly and concisely.
9. Please verify eq. 3.
10. Please complete author contribution or remove.
11. Please complete acknowledgement.
12. The figures have good quality.
14. The Conclusion is OK.
15. Please verify journal abbreviation and respect author guide.
Moderate revision
Author Response
Response to Reviewer 1
Point 1: The topic is of general interest, and the presentation is clear.
Point 2: This article contains new aspects regarding the immobilization of heavy metals.
Point 3: In manuscript, the authors underline the major findings of their work and explain how the use of their proposed procedures and materials represents a progress comparatively with other researches.
Response of Point 1, 2 and 3: Thanks very much for your positive comments. It is our pleasure to receive your comments.
Point 4: In Abstract please rewrite the phrases: “and the other physical properties meet….” and “and the immobilization of Zn; Pb; As; Ni and Cd has a great relation with them”.
Response 4: Now we have re-written this part according to the reviewer’s suggestion. We copy them here for your check:
The results show that the compressive strength of the geopolymer matrix reaches 15.32 MPa after curing for 28 days under the best parameters, and the physical properties meet the requirement of MU10 grade fly ash brick. (Lines 17-19)
The results show that the geopolymer gel and Friedel’s salt are the main hydration products. The S/S mechanism of the arsenic and heavy metals in the geopolymer matrix mainly involves physical encapsulation of the geopolymer gel, geopolymer adsorption, and ion exchange of Friedel’s salt. (Lines 23-26)
Point 5: The key words permit found article in the current registers or indexes.
Point 6: The Introduction reflects state of the art.
Point 7: The text is easy to understand by scientists in other disciplines.
Point 8: The text is presented and arranged clearly and concisely.
Response of Point 5, 6, 7 and 8: Thanks very much for your comments again. It is our pleasure to receive your positive comments.
Point 9: Please verify eq. 3.
Response 9: We have re-written the eq. 3 in our manuscript. We copy them here for your check:
M + Si(OH)4- + Al(OH)4- Mn-[-SiO2-AlO2]n∙wH2O (Geoploymer)
Where M is a cation, usually a Na+, K+ or Ca2+, n is a degree of polycondensation, w≤3 and z is 1, 2 or 3. (Lines 167-169)
Point 10: Please complete author contribution or remove.
Response 10: We have removed the author contribution.
Point 11: Please complete acknowledgement.
Response 11: We have completed the acknowledgement. We copy them here for your check:
The authors gratefully acknowledge the National Key R&D Program of China, the key project of the National Natural Science Foundation of China, the Natural Science Foundation of China, and the Science and Technology Project of Hunan Province for financial support. (Lines 295-297)
Point 12: The figures have good quality.
Point 14: The Conclusion is OK.
Response of Point 12-14: Thanks for your positive comments.
Point 15: Please verify journal abbreviation and respect author guide.
Response 15: We have re-written the references according to the ‘Quick Reference Formatting Guide’ of the IJERPH. (Lines 300-520)

Reviewer 2 Report
Line 104: the Si content very low for an ash. Could you elaborate why the content is so low? The sum of element concentrations is ~55 wt.%. Is rest oxygen?
Line 116: Nice to see that you report sodium silicate information properly (as often it is not very well reported)
Line 190-191: Could elaborate why the band shift would mean that heavy metal was intruded to into Friedel's salt?
Lines 206-210: Why is this information relevant for this paper? Maybe leave out.
Table 3. Very good results, particularly when stabilizing arsenic which is often really difficult to trap in alkaline conditions. Maybe add some comments to the text that S/S of anionic species is often difficult in geopolymers, but Friedel's salt seems to S/S even them. Did you analyse the leaching of other heavy metals present in the precursors (in table 1 and 2)?
Author Response
Response to Reviewer 2
Point 1: Line 104: The Si content very low for an ash. Could you elaborate why the content is so low? The sum of element concentrations is ~55 wt.%. Is rest oxygen?
Response 1: According to Ran Fan et al., study, the Si mainly existed in the form of SiO2, Ca2SiO4 and Na(AlSi3O8) in MSWI bottom ash. Therefore, the Si content of the MSWI fly ash was very low. (Ran Fan et al., Principal component analysis of the two incineration slag. Research and exploration in laboratory. 2014. 33(2). 18-21)
The MSWI fly ash was also detected by XRF, and the elemental compositions result of the MSWI fly ash was listed below.
Element | Fe | O | Si | Cl | Al | Ca | Mg | K | Na | S | Zn | Pb | Cu | Cr | Ti | P |
MSWI FA | 1.28 | 24.59 | 2.62 | 17.98 | 1.07 | 39.45 | 1.23 | 3.27 | 3.25 | 3.11 | 0.92 | 0.21 | 0.07 | 0.02 | 0.34 | 0.20 |
Besides, the XRD analysis (Fig. 2) shown that there were plenty of CaCO3 and Ca(OH)2 in MSWI fly ash. Compared with the result of chemical analysis by ICP-OES method and combined with the results of XRD analysis, the mainly rest elements of MSWI fly ash were O, C and H.
Point 2: Line 116: Nice to see that you report sodium silicate information properly (as often it is not very well reported)
Response 2: Thanks very much for your comments. It is our pleasure to receive your comments.
Point 3: Line 190-191: Could elaborate why the band shift would mean that heavy metal was intruded to into Friedel's salt?
Response 3: The band shift is influenced by the induction of electron- withdrawing group and electron-donating group. The band can shift to a higher wave number when the electron-withdrawing group invades the molecular structure. To our knowledge, the PbO2-, CrO42- and AsO43- are electron-withdrawing groups, which may lead to a higher wave number of Al-OH band in Friedel's salt.
Point 4: Lines 206-210: Why is this information relevant for this paper? Maybe leave out.
Response 4: We have removed this information.
Point 5: Table 3. Very good results, particularly when stabilizing arsenic which is often really difficult to trap in alkaline conditions. Maybe add some comments to the text that S/S of anionic species is often difficult in geopolymers, but Friedel's salt seems to S/S even them. Did you analyse the leaching of other heavy metals present in the precursors (in table 1 and 2)?
Response 5: This comment is helpful for improving our paper. We have added the comments to the text. We copy them here for your check:
According to the previous study, the anionic species is often difficult to solidify/stabilize in geopolymers, but Friedel's salt seems to immobilize them effectively. (Lines 73-74)
We also have analyzed the leaching of other heavy metals present in the precursors (in table 1 and 2). Now we have re-written Table 3 according to your suggestions. We copy them here for your check:
Table 3. TCLP test results of the geopolymer matrices cured for 28 days
Element | Zn | Pb | As | Cd | Ni | Cr | Ba | Cu | Ag | Hg | Be | Se |
US EPA Limits | - | 5 | 5 | 1 | - | 5 | 100 | 100 | - | 0.2 | - | - |
L0M100 | <0.01< span=""> | 2.16 | 0.12 | <0.01< span=""> | <0.01< span=""> | 0.02 | 0.85 | <0.01< span=""> | <0.01< span=""> | <0.01< span=""> | <0.01< span=""> | <0.01< span=""> |
L20M80 | <0.01< span=""> | 0.25 | 0.12 | <0.01< span=""> | <0.01< span=""> | 0.02 | 0.68 | <0.01< span=""> | <0.01< span=""> | <0.01< span=""> | <0.01< span=""> | <0.01< span=""> |
L40M60 | <0.01< span=""> | <0.01< span=""> | 0.10 | <0.01< span=""> | <0.01< span=""> | 0.03 | 0.59 | <0.01< span=""> | <0.01< span=""> | <0.01< span=""> | <0.01< span=""> | <0.01< span=""> |
L60M40 | <0.01< span=""> | <0.01< span=""> | <0.01< span=""> | <0.01< span=""> | <0.01< span=""> | <0.01< span=""> | 0.42 | <0.01< span=""> | <0.01< span=""> | <0.01< span=""> | <0.01< span=""> | <0.01< span=""> |
L80M20 | 3.30 | <0.01< span=""> | <0.01< span=""> | 0.1 | 0.1 | <0.01< span=""> | 0.45 | <0.01< span=""> | <0.01< span=""> | <0.01< span=""> | <0.01< span=""> | <0.01< span=""> |
Optimal matrix | <0.01< span=""> | <0.01< span=""> | 0.12 | <0.01< span=""> | <0.01< span=""> | <0.01< span=""> | <0.01< span=""> | <0.01< span=""> | <0.01< span=""> | <0.01< span=""> | <0.01< span=""> | <0.01< span=""> |
